# Three-Dimensional Au(NiMo)/Ti Catalysts for Efficient Hydrogen Evolution Reaction

**DOI:** 10.3390/ma15227901

**Published:** 2022-11-09

**Authors:** Sukomol Barua, Aldona Balčiūnaitė, Jūrate Vaičiūnienė, Loreta Tamašauskaitė-Tamašiūnaitė, Eugenijus Norkus

**Affiliations:** Department of Catalysis, Center for Physical Sciences and Technology (FTMC), LT-10257 Vilnius, Lithuania

**Keywords:** gold, nickel, molybdenum, electrodeposition, galvanic displacement, hydrogen evolution reaction

## Abstract

In this study, NiMo catalysts that have different metal loadings in the range of ca. 28–106 µg cm^−2^ were electrodeposited on the Ti substrate followed by their decoration with a very low amount of Au-crystallites in the range of ca. 1–5 µg cm^−2^ using the galvanic displacement method. The catalytic performance for hydrogen evolution reaction (HER) was evaluated on the NiMo/Ti and Au(NiMo)/Ti catalysts in an alkaline medium. It was found that among the investigated NiMo/Ti and Au(NiMo)/Ti catalysts, the Au(NiMo)/Ti-3 catalyst with the Au loading of 5.2 µg cm^−2^ gives the lowest overpotential of 252 mV for the HER to reach a current density of 10 mA·cm^−2^. The current densities for HER increase ca. 1.1–2.7 and ca. 1.1–2.2 times on the NiMo/Ti and Au(NiMo)/Ti catalysts, respectively, at −0.424 V, with an increase in temperature from 25 °C to 75 °C.

## 1. Introduction

Although fossil fuels, such as coal, oil, and natural gas, are the main energy sources and are widely used to meet energy needs, the increasing emissions of pollutants, carbon dioxide (CO_2_), and other greenhouse gases require the development of sustainable technologies to meet ever-increasing energy needs. Among various candidates to fulfill energy demands, hydrogen (H_2_) can be a potential substitute fuel for effective energy production and storage. H_2_ is a clean, economical renewable energy source and an excellent energy storage medium with excellent energy conversion efficiency, higher gravimetric energy density than gasoline (120 vs. 44 MJ kg^−1^), eco-friendliness, and zero carbon dioxide emission [1,2,3,4]. H_2_, an important chemical feedstock widely used in petroleum refining and ammonia synthesis, is industrially produced via coal gasification and steam reforming reaction under harsh conditions, resulting in the emission of greenhouse gases and micro-pollutants [5,6]. Among the various available methods, electrocatalytic water splitting is one of the most promising alternatives for H_2_ production that gained intense research interest in the last decades, as electricity-driven water splitting generates green H_2_.

It is well-known that H_2_ production by electrocatalytic water splitting in alkaline media is limited by the sluggish hydrogen evolution reaction (HER) kinetics and enormous electricity consumption. The HER mechanism of electrocatalytic water splitting includes three main steps, i.e., Volmer, Heyrovsky, and Tafel reactions, as shown below, in alkaline media [7]:Volmer reaction: H_2_O + e^−^ → H* + OH^−^(1)
Heyrovsky reaction: H* + H_2_O+ e−→ H_2_ + OH^−^(2)
Tafel reaction: H* + H* → H_2_(3)
Overall reaction: 2H_2_O + 2e^−^ → H_2_ + 2OH^−^(4)

The sluggish HER kinetics in alkaline media is mainly due to the fact that in the Volmer reaction, the proton source comes from the water molecule instead of H_3_O^+^ in the acid electrolyte, which involves additional energy to break the H–O–H bond [7]. To date, platinum (Pt) is the most effective and benchmark electrocatalyst for HER to achieve the lowest overpotential in both acidic and alkaline mediums, but unfortunately, because of the high production costs and the scarcity of resources, the use of Pt or other noble metals for electrodes for the water splitting process is not economically feasible [8,9]. In this context, it is the pursuit of most researchers to find an efficient, cost-effective and stable non-noble metal electrocatalyst for alkaline media to accelerate the Volmer step. Recent studies demonstrated that a number of non-noble transition metal-based materials, including nickel, molybdenum [10,11,12,13,14,15,16], cobalt [17,18,19], iron [20,21], tungsten [22,23], and transition metal compounds (TMCs), such as carbides [23,24,25], phosphide [26,27,28], nitrides [29], and sulfides [30], were investigated as electrocatalysts for HER to achieve excellent chemical stability due to their low cost and sufficient corrosion resistance under alkaline media. Additionally, low-cost transition metals, especially nickel-based electrocatalysts, received noticeable attention as supercapacitor electrodes and bifunctional electrocatalysts due to their abundant reserves, intrinsic high catalytic activity, excellent corrosion resistance, and high electrical conductivity [31]. A number of methods, such as spontaneous galvanic displacement [32], electrochemical deposition [13,33,34], hydrothermal synthesis, etc., were developed to explore and enhance the HER activity of Ni-based bi- and tri-metallic alloy catalysts Ni-M (M = Fe, Co, Mn, Mo, Cr, etc.) and Ni-M bimetallic oxides (BOs).

According to Engel–Brewer valence bond theory, whenever metals of the left half of the transition series (such as Ni and Co) are alloyed with metals of the right half of the transition series metals (Mo or, W), a synergistic effect can be anticipated in terms of hydrogen evolution activity [35]. The synergistic effect between Mo and Ni in the effect of hydrogen binding energy (HBE) is noteworthy, as the HBE between Ni and H is slightly weaker, whereas it is stronger enough between Mo and H. Therefore, the HBE can be controlled to a relatively moderate value by chemically coupling Ni and Mo, which can contribute to balancing the thermodynamics between hydrogen adsorption and desorption [36]. Moreover, enhanced HER activity demonstrated by a self-supported Ni–Mo–P ternary alloy coating on a three-dimensional (3D) Ni foam substrate (Ni–Mo–P/NF) were reported at a current density of −10 mA·cm^−2^ at a small overpotential of −63 mV in 1 M KOH electrolyte [13]. Heterostructured Ni–Mo–N composite nanoparticles, decorated on nitrogen-doped reduced graphene oxide (Ni–Mo–N/NG), also reported an excellent HER electrocatalytic activity with zero onset potential and 46.6 and 159.8 mV overpotentials for 10 and 100 mA·cm^−2^, respectively, in 1 M potassium hydroxide (KOH) solution [14].

This study presents a simple and low-cost procedure to fabricate efficient catalysts for HER. Three-dimensional (3D) binary Ni-Mo catalysts with different total metal loadings supported on a titanium (Ti) surface (denoted as NiMo/Ti) were prepared via the electrodeposition method, whereas for the decoration of the prepared NiMo/Ti catalysts with a small amount of Au crystallites, the galvanic displacement method was used.

## 2. Materials and Methods

### 2.1. Chemicals

Titanium foil (99.7% purity) and HAuCl_4_ (99.995%) were purchased from Sigma-Aldrich (Saint Louis, MO, USA) Supply. H_2_SO_4_ (96%), HCl (35–38%), nickel sulfate hexahydrate (NiSO_4_·6H_2_O, >98%), sodium molybdate dihydrate (Na_2_MoO_4_·2H_2_O, >99.5%), and NaOH (98.8%) were purchased from Chempur Company (Karlsruhe, Germany). Ultra-pure water with a resistivity of 18.2 MΩ·cm^−1^ was used for preparing the solutions. All chemicals were of analytical grade and used directly without further purification.

### 2.2. Fabrication of Catalysts

The catalysts were prepared by a facile, two-step process that involves electrodeposition of Ni^2+^ and Mo^6+^ ion on the surface of the Ti electrode, followed by a spontaneous Au displacement from the Au (III)-containing solution. Before the deposition of the NiMo catalysts, the Ti plates were pretreated in diluted H_2_SO_4_ (1:1 vol) at 70 °C for 3 s. NiMo catalysts were electroplated on the Ti surface (1 × 1 cm) from a bath containing 0.03 M Na_2_MoO_4_ along with 0.1, 0.2, and 1.0 M NiSO_4_ in an acidic condition (1.5 M H_2_SO_4_ and 1 M HCl). The chronopotentiometry was used for the NiMo coatings deposition on the Ti surface. The plating of coatings was carried out at the current of 0.1 mA and 1 mA for 3 min at each current. The Au crystallites were deposited on the prepared NiMo/Ti electrodes by their immersion into 1 mM HAuCl_4_ + 0.1 M HCl solution for 10 s. After plating, the samples were taken out, thoroughly rinsed with deionized water, and air-dried at room temperature.

### 2.3. Characterization of Catalysts

The morphology and composition of the prepared catalysts were investigated by scanning electron microscopy (SEM) TM 4000 Plus (HITACHI, Tokyo, Japan).

XRD patterns of pure Ti sheet, Ni-Mo/Ti, and Au-Ni-Mo/Ti catalysts were measured using an X-ray diffractometer D2 PHASER (Bruker, Karlsruhe, Germany). The measurements were conducted in the 2θ range of 10–90°.

The metal loadings were determined by inductively coupled plasma optical emission spectrometry (ICP–OES) analysis. The ICP–OES spectra were recorded using an Optima 7000DV spectrometer (Perkin Elmer, Waltham, MA, USA) at wavelengths of λ_Ni_ 231.604 nm, λ_Mo_ 202.031 nm, and λ_Au_ 267.595 nm.

### 2.4. Electrochemical Measurements

A conventional three-electrode electrochemical cell was used for electrochemical measurements. The fabricated NiMo/Ti and Au(NiMo)/Ti catalysts were employed as working electrodes, a Pt sheet was used as a counter electrode, and a calomel electrode was used as a reference. All potentials in this work were converted to the reversible hydrogen electrode (RHE) scale using the following Equation (5):*E*_RHE_ = *E*_SCE_ + 0.242 V + 0.059 V × pH_solution._(5)

Current densities were calculated using the electrodes’ geometric area of 2 cm^2^. Linear sweep voltammograms were recorded in a 1 M NaOH solution and always deaerated by argon (Ar) for 20 min prior to measurements. HER polarization curves were recorded from the open circuit potential (OCP) to −0.42 V (vs. RHE) at a polarization rate of 10 mV·s^−1^. Polarization curves were recorded at several temperatures from 25 to 75 °C, and temperatures were set with a water jacket connected to a LAUDA Alpha RA 8 thermostat. Stability was studied by recording chronoamperometry (CA) curves for HER at a potential of −0.22 V (vs. RHE) for half an hour. All electrochemical measurements were performed with a Metrohm Autolab potentiostat (PGSTAT302, Utrecht, The Netherlands) using the Electrochemical Software (Nova 2.1.4).

## 3. Results

In this study, we investigated the electrocatalytic activity of prepared 3D binary NiMo/Ti and ternary Au(NiMo)/Ti catalysts for HER. These catalysts were deposited on the Ti surface (1 × 1 cm) using an electrochemical bath. The optimal conditions for different 3D binary catalyst depositions were determined and are given in Table 1. The electrochemical deposition was carried out by applying the constant current of 0.1 mA and 1 mA for 3 min at each current. The Au crystallites were deposited on the prepared NiMo/Ti electrodes by their immersion into 1 mM HAuCl_4_ + 0.1 M HCl solution for 10 s at room temperature.

The morphology and composition of the prepared catalysts were studied by SEM. Figure 1 shows SEM images of the prepared different compositions 3D NiMo/Ti and Au(NiMo)/Ti catalysts. The low magnification SEM image of the 3D NiMo/Ti-3 catalyst (Figure 1c) proves the formation of many cedar leaf-like Ni-Mo alloy structures in a large area, in which leaf-like Ni-Mo alloy is uniformly dispersed on the Ti foil. These cedar leaf-like structures still retain much space among the leaves, forming a porous morphology that can be expected to facilitate electrolyte penetration. Many nanoparticles can be seen in Figure 1c, and they irregularly stack together, forming a cedar leaf-like structure. When Au crystallites were deposited on the prepared NiMo/Ti-3 electrode by being immersed into 1 mM HAuCl_4_ + 0.1 M HCl solution for 10 s, the porous leaf-like alloy structure was immediately covered with a tiny globular surface (Figure 1f). The mass of the elements (metal loadings) on the Ti substrate surface was determined by ICP–OES analysis (Table 2).

It can be seen that the formed 3D binary NiMo/Ti catalysts contained ca. 82.8–93.7 wt.% of Ni, whereas those 3D ternaries Au(NiMo)/Ti catalysts possessed 76.6–87.9 wt.% of Ni. The total metal loadings (µg_metal_cm^−2^) in the prepared catalysts are quite different and vary from 23.9 up to 106.2 µg_metal_cm^−2^. It should be noted that Ni and Mo amounts increase in the NiMo coatings by increasing the Ni^2+^ concentration in the plating solution, whereas the Mo amount was kept the same. Calculated Mo:Ni and Au:NiMo mass weight ratios are given in Table 2. 

As seen from the data in Table 2, mass weight ratios Mo:Ni for NiMo/Ti increase due to the rise of Ni^2+^ concentration in the plating solution. A similar phenomenon is observed in the case of Au crystallite-modified NiMo/Ti catalysts. Mass ratios Mo:Ni also increase with the increase in the Ni^2+^ concentration in the plating solution (Table 2). Moreover, Au loadings in the AuNiMo/Ti-1, AuNiMo/Ti-2, and AuNiMo/Ti-3 catalysts increased while the deposition times of Au crystallites were the same—10 s. The mass ratio Au:NiMo slightly decreases. The increased Ni amount in the catalysts allows for achieving a higher Au loading in the ones.

Figure 2 shows XRD patterns for a pure Ti sheet (lower curve) and NiMo/Ti-3 and Au(NiMo)/Ti-3 catalysts (upper curves). Symbols indicate the positions of the XRD peaks of Ti (ICDD card no 00-044-1294).

The lowest XRD pattern (a) in Figure 2 contains sharp XRD peaks of the Ti sheet corresponding to the hexagonal structure of Ti. In the case of NiMo/Ti-3 and Au(NiMo)/Ti-3 catalysts, XRD peaks corresponding to Mo (110) and Mo (200) are shifted towards higher diffraction angles with respect to the positions of Mo presented in ICDD card no 00-044-1120. Additionally, the body-centered cubic lattice parameter decreases from 3.147 to 3.093 Å. This is the result of the formation of a solid solution of Ni (ICDD # 00-004-0850) with a small amount of Mo and Mo-Ni solid solution. There are no visible changes in the XRD patterns for NiMo/Ti-3 and Au(Ni-Mo)/Ti-3 (Figure 2, b,c curves) as the Au (ICDD # 00-004-0784) peaks can be amorphous or crystalline with low intensity.

The electrocatalytic properties of the prepared catalysts were investigated by recording LSVs in 1.0 M NaOH solution at a potential scan rate of 10 mV·s^–1^ in a potential range from open-circuit potential (OCP) up to −0.42 V (vs. RHE) for HER, at several temperatures from 25 up to 75 °C (Figure 3). Ternary Au(NiMo)/Ti-3 coating exhibited the highest current density (j), followed by Au(NiMo)/Ti-2 and Au(NiMo)/Ti-1, and the fabricated binary (NiMo/Ti) catalysts exhibited notably lower current density, in mutual comparison for HER (Figure 3). For those binary NiMo/Ti catalysts, the current density increases ca. 1.2–2.7 times with an increase in temperature from 25 up to 75 °C, whereas fabricated 3D ternary Au(NiMo)/Ti catalysts exhibit ca. 1.1–2.2 times higher current density for HER.

For instance, the current densities of −49.84, −40.73, and −36.58 mA·cm^−2^ were reached at −0.424 V (vs. RHE) using Au-decorated ternary Au(NiMo)/Ti-3, Au(NiMo)/Ti-2, and Au(NiMo)/Ti-1 catalysts, and relatively lower −34.81, −26.5, and −21.75 mA·cm^−2^ current densities were recorded at the same potential via using 3D binary NiMo/Ti-3, NiMo/Ti-2, and NiMo/Ti-1 catalysts at 25 °C, respectively (Figure 4a,b, Table 3). Overpotentials (vs. RHE) to reach the current density of 10 mA·cm^−2^ (η_10_) were found in a gradual increasing order for both Au(NiMo)/Ti and NiMo/Ti catalysts as follows:

Au(NiMo)/Ti-3 (−252 mV) < Au(NiMo)/Ti-2 (−298 mV) < Au(NiMo)/Ti-1 (−308 mV)

NiMo/Ti-3 (−288 mV) < NiMo/Ti-2 (−344 mV) < NiMo/Ti-1 (−349 mV).

It was determined that mass weight ratios Mo:Ni for NiMo/Ti catalysts increase due to the increased Ni^2+^ concentration in the plating solution (Table 2). A higher mass–weight ratio Mo:Ni induces a more pronounced activity of the NiMo/Ti-3 catalyst for HER. Additionally, the increased amount of Ni in the NiMo/Ti catalysts allows for achieving higher Au loading in the Au(NiMo)/Ti catalysts. This is the main factor that influences the lowering overpotential at Au(NiMo)/Ti catalysts compared with NiMo/Ti catalysts. The higher activity of Au crystallite-modified NiMo/Ti catalysts may be related with the synergetic effect of Au, Ni, and Mo [35].

HER polarization curves were then further used for constructing the Tafel plots and calculating the Tafel slope. Tafel slope values of 99.6, 100.5, and 130.4 mV·dec^−1^ were found for HER at NiMo/Ti-1, NiMo/Ti-2, and NiMo/Ti-3 catalysts, respectively. For those 3D ternary Au(NiMo)/Ti catalysts, Tafel slope values of 143.8, 98.7, and 131.2 mV·dec^−1^ were determined at Au(NiMo)/Ti-1, Au(NiMo)/Ti-2, and Au(NiMo)/Ti-3 catalysts, respectively (Figure 4a’,b’, Table 3). The determination of the Tafel slope explores the HER kinetics by measuring the increase in current density with the increase in overpotential, whereas the exchange current density (j_0_) reflects the electrode’s intrinsic activity for HER. The exchange current density (j_0_) was calculated for HER at all six catalysts by extrapolating the Tafel plots, η vs. log j. Thus, the j_0_ value of 0.144, 0.011, 0.076, 0.075, 0.006, and 0.005 mA·cm^−2^ were calculated for Au(NiMo)/Ti-3, Au(NiMo)/Ti-2, Au(NiMo)/Ti-1, NiMo/Ti-3, NiMo/Ti-2, and NiMo/Ti-1 catalysts, respectively (Table 3). It is worth noting that the j_0_ value determined for HER at the Au(NiMo)/Ti-3 coating was ca. 2–28 times higher than that determined for the rest of the studied catalysts.

Another crucial criterion for an advanced electrode material is its electrochemical stability. Chronoamperometric measurements with all six catalysts were carried out in 1 M NaOH at −0.22 V for 2 h. Initially, in the first 50–200 s, a decrease in current density was observed for all investigated catalysts. However, after approximately 500 s, the current densities settled down and remained apparently parallel throughout the experiments. CA results confirm the result of LSV analysis in terms of the ternary Au(NiMo)/Ti-3 catalyst, giving the highest current density during HER (−10.36 mA·cm^−2^ at 50 s) (Figure 5). A more than 2.5 times lower current density was obtained with Au(NiMo)/Ti-2 (−3.81 mA·cm^−2^) and ca. 5 times lower with Au(NiMo)/Ti-1 catalysts (−2.02 mA·cm^−2^). In the case of the binary NiMo/Ti-3 catalyst, a comparatively lower current density was recorded (−6.13 mA·cm^−2^ at 50 s) along with a 3–5 times lower value for NiMo/Ti-2 (−2.06 mA·cm^−2^) and NiMo/Ti-1 (−1.23 mA·cm^−2^) catalysts.

A comparison of HER parameters generated using herein-tested NiMo/Ti and Au(NiMo)/Ti catalysts in an alkaline medium with some electrodes reported in the literature is given in Table 4.

## 4. Conclusions

In summary, NiMo and Au(NiMo) catalysts supported on a titanium surface were studied as electrocatalysts for HER in an alkaline medium. NiMo/Ti catalysts with different total metal loadings in the range of ca. 28–106 µg cm^−2^ were prepared using a simple and low-cost metal electrodeposition method. The decoration of the prepared NiMo/Ti catalysts with a small amount of Au-crystallites in the range of ca. 1–5 µg cm^−2^ was carried out using the galvanic displacement method.

It was determined that, among the investigated catalysts, the Au(NiMo)/Ti-3 catalyst with the Au loading of 5.2 µg cm^−2^ exhibits the highest current density, as well as exchange current density during HER in a 1 M NaOH solution. Moreover, the Au(NiMo)/Ti-3 catalyst also displays excellent HER performance with an overpotential of 252 mV at a current density of 10 mA·cm^−2^.

## Figures and Tables

**Figure 1 materials-15-07901-f001:**
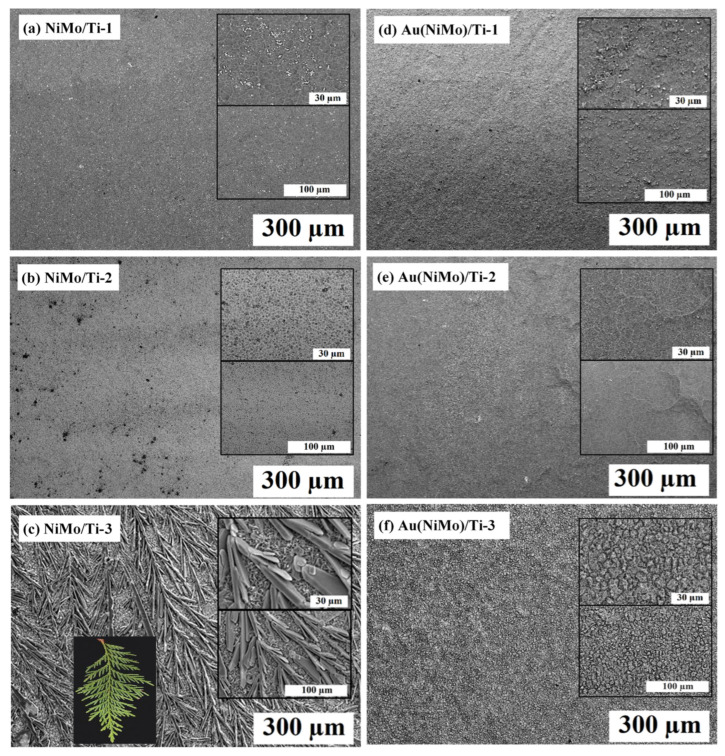
SEM views of NiMo/Ti (**a**–**c**) and Au(NiMo)/Ti (**d**–**f**) catalysts mentioned in Table 2. (**c**) The inset represents a photo of a cedar leaf.

**Figure 2 materials-15-07901-f002:**
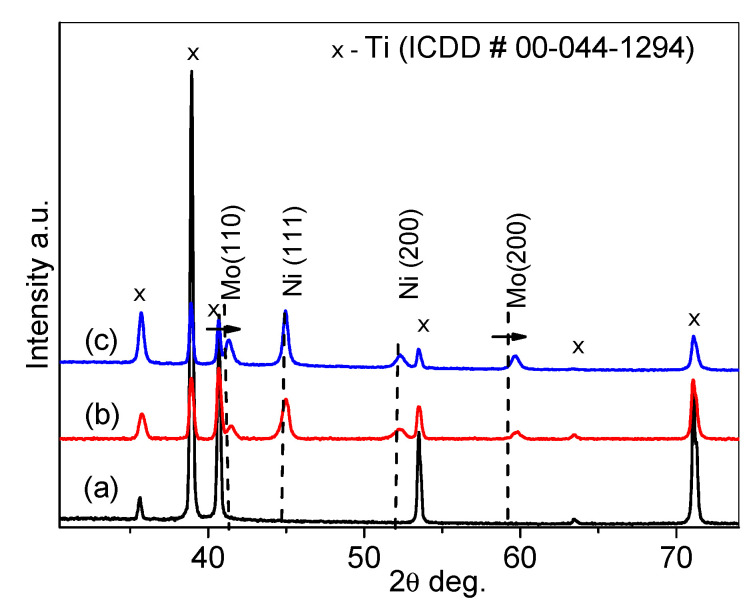
XRD patterns of Ti sheet (a), Ni-Mo/Ti-3 (b), and Au(Ni-Mo)/Ti-3 (c) catalysts.

**Figure 3 materials-15-07901-f003:**
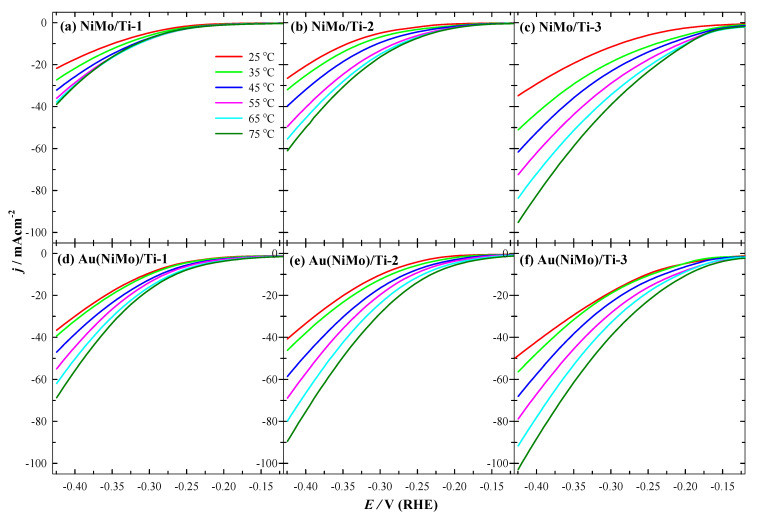
HER polarization curves of 3D NiMo/Ti (**a**–**c**) and Au(NiMo)/Ti (**d**–**f**) catalysts in 1 M NaOH solution at a 10 mV·s^−1^ potential scan rate and a temperature range (25–75 °C).

**Figure 4 materials-15-07901-f004:**
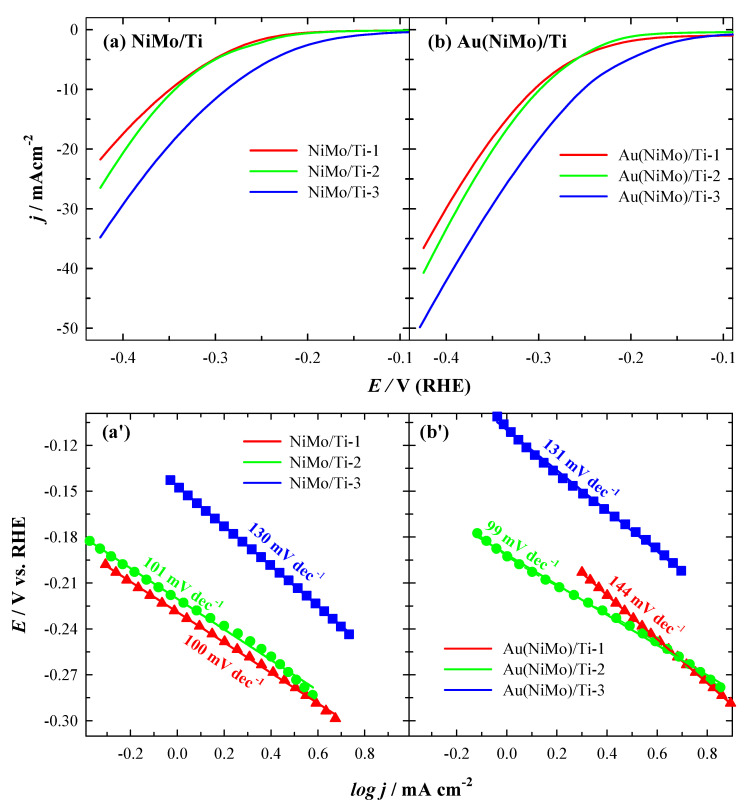
HER polarization curves of 3D NiMo/Ti (**a**) and Au(NiMo)/Ti (**b**) catalysts in 1 M NaOH solution at a potential scan rate of 10 mV·s^−1^ and 25 °C temperature and corresponding Tafel plots (**a’**,**b’**).

**Figure 5 materials-15-07901-f005:**
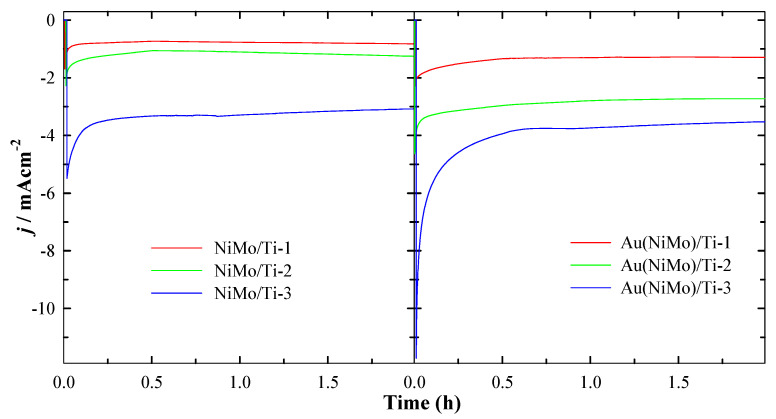
Chronoamperometric data of the investigated NiMo/Ti and Au(NiMo)/Ti catalysts in 1 M NaOH solution at the potential value of −0.22V (vs. RHE), *t* = 2 h.

**Table 1 materials-15-07901-t001:** Composition of the electrochemical bath.

Catalysts	Concentration in Mol dm^−3^	
	Ni^2+^	Mo^6+^
NiMo/Ti-1	0.1	0.03
NiMo/Ti-2	0.2	0.03
NiMo/Ti-3	1.0	0.03

**Table 2 materials-15-07901-t002:** The metal loading in the catalysts was determined by ICP-OES analysis and metal mass weight ratio.

Catalyst	Ni Loadings (µg_Ni_cm^−2^)	Mo Loadings (µg_Mo_cm^−2^)	Auloadings (µg_Au_cm^−2^)	Total Metal Loading (µg_metal_cm^−2^)	Mass Weight Ratio	
					Mo:Ni	Au:NiMo
NiMo/Ti-1	23.4	4.9		28.3	1:4.78	
NiMo/Ti-2	29.6	5.3		34.9	1:5.58	
NiMo/Ti-3	99.5	6.7		106.2	1:14.85	
Au(NiMo)/Ti-1	18.3	4.4	1.2	23.9	1:4.16	1:18.92
Au(NiMo)/Ti-2	25.4	4.6	1.7	31.7	1:5.52	1:17.65
Au(NiMo)/Ti-3	81.4	6.0	5.2	92.6	1:13.57	1:16.81

**Table 3 materials-15-07901-t003:** Electrochemical performance of the tested catalysts toward HER in alkaline media.

Catalysts	*j* (mA·cm^−2^) in Different Temperatures (°C) at −0.424 V						Tafel Slope (mV·dec^−1^)	η_10_ * (mV)	j_0_ (mA·cm^−2^)
	25	35	45	55	65	75			
NiMo/Ti-1	−21.75	−27.25	−32.14	−35.99	−37.83	−38.91	99.6	−349	0.005
NiMo/Ti-2	−26.5	−31.94	−39.72	−49.53	−55.44	−61.05	100.5	−344	0.006
NiMo/Ti-3	−34.81	−51.09	−61.63	−72.42	−83.62	−95.19	130.4	−288	0.075
Au(NiMo)/Ti-1	−36.58	−39.32	−47.04	−54.93	−61.86	−68.68	143.8	−308	0.076
Au(NiMo)/Ti-2	−40.73	−46.19	−58.55	−68.86	−79.8	−89.45	98.7	−298	0.011
Au(NiMo)/Ti-3	−49.84	−56.38	−68	−78.81	−91.6	−102.86	131.2	−252	0.144

* Overpotential at 10 mA cm^−2^.

**Table 4 materials-15-07901-t004:** The electrochemical performance of herein tested catalysts towards HER in alkaline media and compared with that of transition metal-based electrodes reported in the literature.

Catalyst	Overpotential η_10_ ** (mV)	Tafel Slope (mV·dec^−1^)	Temperature (°C)	Electrolyte	Ref.
Au(NiMo)/Ti-3	−252	131.2	25	1 M NaOH	This work
NiMo/Ti-3	−288	130.4	25	1 M NaOH	This work
Au(NiMo)/Ti-2	−298	98.7	25	1 M NaOH	This work
NiMo/Ti-2	−344	100.5	25	1 M NaOH	This work
Au(NiMo)/Ti-1	−308	143.8	25	1 M NaOH	This work
NiMo/Ti-1	−349	99.6	25	1 M NaOH	This work
Ni-Mo-O MCFs	−222.8	141.6	-	1 M KOH	[33]
NiFeCMo-30	−254	163.9	-	30% KOH	[34]
NiS_2_/MoS_2_ HNW	−204	65	-	1 M KOH	[37]
Ni–Cr–Mo–Fe, Ni–Cr–Mo, Ni–Cr alloy	−232 −249 −255	57.7 61.1 62.3	25	1 M KOH	[38]
Ni–Mo/WC 1, Ni–Mo/WC 2, Ni–Mo/WC 3	−411 −262 −134	208 153 163	25	1 M KOH	[39]
Ni/TM-360 s	−205	104	-	1 M KOH	[40]
NiCu_0.57_/Ni_3_S_2_/TM, Ni/Ni_3_S_2_/TM	−239 −441	86 195	-	1 M KOH	[41]
Ni_3_Te_2_-Ni foam Ni_3_Te_2_-Au glass Ni_3_Te_2_-Hydrothermal	−212 −237 −304	126.2 73.1 94.2	-	1 M KOH	[42]
NiTe_2_-nanosheet	−256	98	-	1 M KOH	[43]
NiTeNR/NF	−248	185	-	1 M KOH	[44]
NiTe_2_	−520	188.3	-	1 M KOH	[45]

MCFs—mesoporous composite films, HNW—hybrid nanowire, WC—tungsten carbide, TM—Ti mesh, and NR/NF—nanorods/Ni foam. ** Overpotential at 10 mA·cm^−2^.

## Data Availability

Not applicable.

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
