# Peer review of "Three-Dimensional Au(NiMo)/Ti Catalysts for Efficient Hydrogen Evolution Reaction"

_materials, 2022, doi:10.3390/ma15227901_

Round 1

Reviewer 1 Report

I recommended to publish this paper after further work experiment such as XRD study to the prepared samples and Gas chromatography to determine the hydrogen evolution rate

Author Response

Author's Reply to the Review Report (Reviewer 1)

I recommended to publish this paper after further work experiment such as XRD study to the prepared samples and Gas chromatography to determine the hydrogen evolution rate.

Thank you, we do not have the possibility to do XRD and gas chromatography measurements at the moment.

Reviewer 2 Report

The paper by Barua et al. reports nice and original results on the three-dimensional Au(NiMo)/Ti catalysts for efficient hydrogen evolution reaction. The paper is interesting and its impact is fully adequate for Materials.

The reviewer recommends publication pending minor revisions:

1) it would be better to show the chronoamperometric curves in a timescale of hours rather than seconds to help the reader

2) Refs. 6 and 32 should be replaced. Ref. 6 should be replaced by a more general reference on H2-based technology. 

3) considering the comparison with Ni/Ni3S2, it would be good also to compare with NiTe2 and NiSe, i.e., Ni-based chalcogenides.

Author Response

Author's Reply to the Review Report (Reviewer 2)

Comments and Suggestions for Authors

The paper by Barua et al. reports nice and original results on the three-dimensional Au(NiMo)/Ti catalysts for efficient hydrogen evolution reaction. The paper is interesting and its impact is fully adequate for Materials.

The reviewer recommends publication pending minor revisions:

1) it would be better to show the chronoamperometric curves in a timescale of hours rather than seconds to help the reader

Thank you, we show the chronoamperometric curves in a timescale of hours.

2) Refs. 6 and 32 should be replaced. Ref. 6 should be replaced by a more general reference on H2-based technology. 

Thank you, we replaced references in the revised manuscript.

3) considering the comparison with Ni/Ni3S2, it would be good also to compare with NiTe2 and NiSe, i.e., Ni-based chalcogenides.

Thank you, the comparison was added in the revised manuscript.

Reviewer 3 Report

Sukomol Barua et al. developed Au(NiMo)-based catalysts on a titanium surface using electroplating and galvanic displacement techniques and studied three-dimensional Au(NiMo)/Ti catalysts catalytic activity for hydrogen evolution reaction. However, some serious issues need to be addressed in the manuscript.

Ø Authors mentioned in the manuscript that the process of electrochemical deposition was carried out at the current of 0.1 mA and 1 mA for 3 min at each current. What electrochemical technique is used need to be mentioned.

Ø The Au crystallites were deposited on the prepared NiMo/Ti electrodes by their immersion into 1 mM HAuCl4 + 0.1 M HCl solution for 10 s. Is it possible in 10 s?

Ø The XRD characterization has not been carried out to know the phase of the synthesized materials which is one of the most essential and important characterization.

Ø The HER catalytic studies has been carried out in 1 M NaOH solution, why authors did not use KOH? Most of the researchers used KOH for HER studies. For uniform comparison it is good to use KOH solution.

Ø Generally, in alkaline medium Hg/HgO reference electrode is recommended to use.

Ø After synthesized materials, in what ration Au, Ni and Mo are present, EDAX characterization or ICP analysis is required for that.

Ø  The obtained overpotentals are relatively high compared to the recent reports which is not good and Au is expensive to use.

Ø Authors mentioned in characterization of catalysts section that EDAX and ICP analysis carried out but not given in the manuscript.

Ø Authors studied stability studied using chronoamperometry for only 30 min (1800 sec), which is not acceptable at least for 20-30 h is required.

Ø Throughout the manuscript only one SEM physical characterization is given in the manuscript.

Ø Tafel plots should be presented in an appropriate way.

Ø Abstract need to be improved, introduction divided into many paragraphs some of them can be merged.

Author Response

Author's Reply to the Review Report (Reviewer 3)

Comments and Suggestions for Authors

Sukomol Barua et al. developed Au(NiMo)-based catalysts on a titanium surface using electroplating and galvanic displacement techniques and studied three-dimensional Au(NiMo)/Ti catalysts catalytic activity for hydrogen evolution reaction. However, some serious issues need to be addressed in the manuscript.

Ø Authors mentioned in the manuscript that the process of electrochemical deposition was carried out at the current of 0.1 mA and 1 mA for 3 min at each current. What electrochemical technique is used need to be mentioned.

Thank you, the chronopotentiometry was used for the electrodeposition.

Ø The Au crystallites were deposited on the prepared NiMo/Ti electrodes by their immersion into 1 mM HAuCl4 + 0.1 M HCl solution for 10 s. Is it possible in 10 s?

Thank you, yes, it is possible. The deposition of Au crystallites was confirmed by ICP-OES analysis.

Ø The XRD characterization has not been carried out to know the phase of the synthesized materials which is one of the most essential and important characterization.

Thank you, we do not have possibility to do XRD measurements at the moment.

Ø The HER catalytic studies has been carried out in 1 M NaOH solution, why authors did not use KOH? Most of the researchers used KOH for HER studies. For uniform comparison it is good to use KOH solution.

Thank you, we usually used 1M NaOH as a background solution for other experiments and we decided to try to do an investigation for HER in the same solution. For the next experiments, we will consider using KOH instead of NaOH for HER.

Ø Generally, in alkaline medium Hg/HgO reference electrode is recommended to use.

Thank you, we do not have possibility to use Hg/HgO reference electrode.

Ø After synthesized materials, in what ration Au, Ni and Mo are present, EDAX characterization or ICP analysis is required for that.

The composition of catalysts was determined by ICP-OES analysis. Additionaly we added the mass weight ratio Mo:Ni for NiMo/Ti catalysts and Au:NiMo for AuNiMo/Ti catalysts in Table 2.

Ø  The obtained overpotentals are relatively high compared to the recent reports which is not good and Au is expensive to use.

Ø Authors mentioned in characterization of catalysts section that EDAX and ICP analysis carried out but not given in the manuscript.

The composition of catalysts was determined by ICP-OES analysis. The obtained data are given in Table 2. Additionally, the mass weight ratio Mo:Ni and Au:NiMo for NiMo/Ti and Au(NiMo)/Ti catalysts, respectively was calculated and given in Table 2.

Ø Authors studied stability studied using chronoamperometry for only 30 min (1800 sec), which is not acceptable at least for 20-30 h is required.

Thank you, we added the stability studies using chronoamperometry for 2 hours.

Ø Throughout the manuscript only one SEM physical characterization is given in the manuscript.

Thank you, we do not have possibility to do XRD or XPS measurements at the moment.

Ø Tafel plots should be presented in an appropriate way.

Tafel plots were checked.

Ø Abstract need to be improved, introduction divided into many paragraphs some of them can be merged.

Reviewer 4 Report

In this paper, the author deposited Ni-Mo alloy coating on Ti substrate by electrodeposition. Then they decorated the material with Au to prove the catalytic performance for hydrogen evolution reaction. The results showed that the incorporation of Au did improve the catalytic performance and Au(NiMo)Ti-3 showed the lowest overpotential. I recommend this article for publication after minor revision.

(1) I'm curious as to why you chose to use Au for decorations. Is there any scientific reason behind it? Have you tried other elements? As we know, Au is much more expensive than Ni. With the addition of Au, is the performance increase significant enough to justify the higher cost in practical applications?

(2)The author mentioned in the introduction that there's synergistic effect between Mo and Ni. Is there any experimental evidence for that? Is there anything worth mentioning in the XPS measurement?

(3)If there is synergistic effect between Mo and Ni, what about between Au, Mo and Ni. What role does Au play in improving the catalytic performance? Can you elaborate more in the manuscript?

(4)As discussed in the manuscript, for the Au decorated catalysts, the overpotential for hydrogen evolution reaction was found in a order of: NiMo/Ti-3<NiMo/Ti-2<NiMo/Ti-1. Comparing this three catalysts, the metal loading for the three elements all increased. So the question is: is the activity increase from the increase of Au,  or the increase of the total metal loading?

(5)For the electrochemical performance of the tested catalysts at different temperatures in Table 3, specifically the Au(NiMo)Ti-2 seems to have a really small Tafel slope, as well as a low j0. This is somewhat not consistent with the trend other data is showing. Is there any explanation for that?

(6) Have you done any material characterization after the long term stability test, such as SEM, XPS? It'd be interesting to compare the XPS results before and after the electrochemical test, as well as the morphology change after the long term stability test.

(7) The stability test only lasted for 1800s, which in my opinion is not long enough. I'd suggest test it for a much longer time.

Author Response

Author's Reply to the Review Report (Reviewer 4)

Comments and Suggestions for Authors

In this paper, the author deposited Ni-Mo alloy coating on Ti substrate by electrodeposition. Then they decorated the material with Au to prove the catalytic performance for hydrogen evolution reaction. The results showed that the incorporation of Au did improve the catalytic performance and Au(NiMo)Ti-3 showed the lowest overpotential. I recommend this article for publication after minor revision.

(1) I'm curious as to why you chose to use Au for decorations. Is there any scientific reason behind it? Have you tried other elements? As we know, Au is much more expensive than Ni. With the addition of Au, is the performance increase significant enough to justify the higher cost in practical applications?

Thank you, we decorated just with Au particles. The low amount of Au on the surface increases the electrocatalytic activity of the catalysts.

(2)The author mentioned in the introduction that there's synergistic effect between Mo and Ni. Is there any experimental evidence for that? Is there anything worth mentioning in the XPS measurement?

The deposition of Au crystallites on the NiMo/Ti catalysts results in the lowering overpotential for HER compared to the NiMo/Ti catalysts. We do not have possibility to do XPS measurements at the moment.

(3) If there is synergistic effect between Mo and Ni, what about between Au, Mo and Ni. What role does Au play in improving the catalytic performance? Can you elaborate more in the manuscript?

It was discussed in the revised version of the manuscript.

(4) As discussed in the manuscript, for the Au decorated catalysts, the overpotential for hydrogen evolution reaction was found in a order of: NiMo/Ti-3<NiMo/Ti-2<NiMo/Ti-1. Comparing this three catalysts, the metal loading for the three elements all increased. So the question is: is the activity increase from the increase of Au,  or the increase of the total metal loading?

Thank you, the activity increased more with the increase of Au loading.

(5) For the electrochemical performance of the tested catalysts at different temperatures in Table 3, specifically the Au(NiMo)Ti-2 seems to have a really small Tafel slope, as well as a low j0. This is somewhat not consistent with the trend other data is showing. Is there any explanation for that?

We have not explanation for this phenomenon.

(6) Have you done any material characterization after the long term stability test, such as SEM, XPS? It'd be interesting to compare the XPS results before and after the electrochemical test, as well as the morphology change after the long term stability test.

Thank you for suggestion. We do not have possibility to do XPS measurements at the moment.

(7) The stability test only lasted for 1800s, which in my opinion is not long enough. I'd suggest test it for a much longer time.

Thank you, we added the stability studies using chronoamperometry for 2 hours.

Round 2

Reviewer 3 Report

This manuscript has not been sufficiently improved.

For raised comments authors mentioned that we do not have possibility to do.

To know the material(phase/structure) formation at least XRD characterization is required. It is preliminary characterization in materials synthesis.

Some comments are unanswered and not improved sufficiently in the revised manuscript.

Author Response

Response to Reviewer # 3

We want to thank the Reviewer for the valuable comments. We tried to clarify our responses in the revised version of the manuscript.

Reviewer: This manuscript has not been sufficiently improved.

For raised comments authors mentioned that we do not have possibility to do.

To know the material(phase/structure) formation at least XRD characterization is required. It is preliminary characterization in materials synthesis.

Some comments are unanswered and not improved sufficiently in the revised manuscript.

Authors: The additional XRD analysis was carried out and its results were discussed in the revised version of the manuscript.

Reviewer: The obtained overpotentials are relatively high compared to the recent reports which is not good and Au is expensive to use.

Authors: The obtained overpotentials of our catalysts have no significant difference compared with the data of other authors (sometimes higher, sometimes lower). It can be noted that modification of NiMo with Au crystallites results in the decrease of overpotential values compared with the catalysts without Au.

Reviewer: Abstract need to be improved, introduction divided into many paragraphs some of them can be merged.

Authors: Abstract and Introduction parts were thoroughly revised.
